# The Proteogenome of Symbiotic *Frankia alni* in *Alnus glutinosa* Nodules

**DOI:** 10.3390/microorganisms10030651

**Published:** 2022-03-18

**Authors:** Petar Pujic, Nicole Alloisio, Guylaine Miotello, Jean Armengaud, Danis Abrouk, Pascale Fournier, Philippe Normand

**Affiliations:** 1Ecologie Microbienne, CNRS, UMR5557, Université Lyon 1, Université de Lyon; INRA, UMR1418, 7330 Villeurbanne, France; nicole.alloisio@orange.fr (N.A.); danis.abrouk@univ-lyon1.fr (D.A.); pascale.fournier@univ-lyon1.fr (P.F.); 2Département Médicaments et Technologies pour la Santé (DMTS), CEA, INRAE, Université Paris-Saclay, SPI, 30200 Bagnols sur Cèze, France; guylaine.miotello@cea.fr (G.M.); jean.armengaud@cea.fr (J.A.)

**Keywords:** omics, nitrogenase, hopanoids, actinorhizae

## Abstract

Omics are the most promising approaches to investigate microbes for which no genetic tools exist such as the nitrogen-fixing symbiotic *Frankia*. A proteogenomic analysis of symbiotic *Frankia alni* was done by comparing those proteins more and less abundant in *Alnus glutinosa* nodules relative to N-replete pure cultures with propionate as the carbon source and ammonium as the nitrogen-source. There were 250 proteins that were significantly overabundant in nodules at a fold change (FC) ≥ 2 threshold, and 1429 with the same characteristics in in vitro nitrogen-replete pure culture. Nitrogenase, SuF (Fe–Su biogenesis) and hopanoid lipids synthesis determinants were the most overabundant proteins in symbiosis. Nitrogenase was found to constitute 3% of all *Frankia* proteins in nodules. Sod (superoxide dismutase) was overabundant, indicating a continued oxidative stress, while Kats (catalase) were not. Several transporters were overabundant including one for dicarboxylates and one for branched amino acids. The present results confirm the centrality of nitrogenase in the actinorhizal symbiosis.

## 1. Introduction

*Frankia* is a genus containing soil actinobacteria that establish a nitrogen-replete root nodule symbiosis with 24 genera and hundreds of species of dicotyledons [1]. The best known of these is *Alnus glutinosa* L., the designated type, which is representative of the genus both for morphological characters and evolutionary rate [2]. 

Hundreds of *Frankia* strains have been isolated from nodules of the different species and classified according to their physiology [3], host specificity [4] or phylogeny [5]. More recently, many species have been described—the first of which was *Frankia alni* [6]—that can nodulate *Alnus* spp. and *Myrica* spp. Its genome has been deciphered and compared to other species [5] and found to contain no symbiosis island, with all genes related to symbiosis present in several unlinked clusters [7].

This symbiosis between *F. alni* and *Alnus* spp. is initiated by Ca-spiking [8], followed by the synthesis of a root-hair-deforming factor [9] that allows the deformation, entrapment of hyphae and penetration into the cortical cells [10]; the formation of a pre-nodule and the emergence of a mature nitrogen-fixing nodule. On the plant side, the mechanisms involved appear similar to those present in legumes, with several elements of the common symbiotic cascade involved, such as the nuclear transcription factor NIN (nodule inception) [11] and the SYMR (symbiosis) kinase [12]. The genomics of symbiotic and non-symbiotic plant phylogenetic neighbors has shown that the common symbiotic determinants were present in all leguminous and actinorhizal lineages and that loss of symbiotic capacity was accompanied by the loss of *NIN* and *RPG* (rhizobium polar growth) [13].

On the microbial side, less is known since no genetic tools could be developed despite several attempts, hampering mutagenesis and complementation studies. The genomes revealed no canonical *nod* genes [5] except in two cl2 strains [14,15] and one cl3 strain [16] where two gene clusters are found, one with *nodAB_1_* and the other with *nodB_2_CIJ*; however, these were not detected by proteomics of *F. soli* exposed to root exudates from compatible *Elaeagnus angustifolia* and incompatible hosts [17] even though the mRNAs had been detected in *Datisca* nodules [14]. Transcriptomics of *Frankia* in *Alnus* nodules [7] has identified *nif* (nitrogenase), *hop* (hopanoid synthesis), *suf* (FeS cluster synthesis) and *hup* (hydrogenase uptake) as upregulated but has shed no light on the molecular dialogue between the two partners. The determinants and structure of the root deforming factor [10], thus, remain elusive. We also know through omics [18] of early rhizospheric interactions that a cellulase/cellulose synthase cluster is upregulated even though *F. alni* cannot grow using glucose as a carbon source, which is evocative of a local weakening of the hair cell wall and a hardening of the hyphal tip to facilitate entry into root tissues. We also know that the auxin PAA is synthesized in nodules and in pure culture—an auxin that when applied onto roots at 10^−5^ M causes emergence of stunted swollen secondary roots that are similar to nodules [19]. It is known that the plant synthesizes peptides that bind to and modify the porosity of vesicles [20] and a non-comparative survey of field nodules has shown nitrogenase, the protein that reduces dinitrogen into ammonium and tricarboxylate cycle (TCA) proteins to be abundant [21].

Other in vitro proteomic studies have been done of *Frankia* with/without ammonium [22] or under osmotic stress [23] and have shown TCA involvement, stress determinants and various regulators.

We undertook the present study to characterize the proteomic response of *Frankia alni* as it forms mature 21 dpi nodules on *A. glutinosa* roots.

## 2. Materials and Methods

### 2.1. Plant and Bacterial Material

*Frankia alni* strain ACN14a [24] cells were maintained in BAP medium [25] with 5 mM propionate as the carbon source and 5 mM NH_4_^+^ as the nitrogen source buffered to pH 6.5.

*Alnus glutinosa* seeds were harvested from a tree growing in Lyon, France, used previously [7]. They were grown as before with some modifications: seedlings were transferred to Fahraeus’s solution [26] in opaque plastic pots (eight seedlings/pot) and grown for four weeks with 0.5 g·L^−1^ KNO_3_, followed by one week without KNO_3_ before inoculation with *F. alni* [24]. To inoculate seedlings, *Frankia* cells were grown in BAP-PCM medium (4 × 250 mL) until log-phase [27]. Cells were collected by centrifugation, washed twice with sterile ultra-pure water and resuspended in 500 mL of Fahraeus’s solution without KNO_3_. Cell cultures were homogenized by syringing though a 21G needle. The *Frankia* cell suspensions were applied onto plant roots (symbiotic condition). After 21 days, mature nodules were harvested and ground in liquid nitrogen.

As a reference, *F. alni* cells were inoculated after syringing with a series of needles (21G, 23G, 25G, 27G) and grown for 10 days (corresponding to the end of the exponential phase) in 250 mL of BAP medium with ammonium (5 mM) in agitated 500 mL Erlenmeyer flasks [25] buffered to pH 6.5. No vesicles could be found.

### 2.2. Proteome Characterization

Each sample was dissolved in LDS 1X buffer (Invitrogen, Carlsbad, CA, USA) with 100 µL of LDS 1X per 30 mg of pellet. The solutions were warmed at 99 °C for 5 min and subjected to sonication in an ultrasonic bath for 5 min. Each sample was transferred into a 2 mL Precellys (Bertin Technologies, Montigny-le-Bretonneux, France) tube containing 200 mg of glass beads and subjected to three cycles of grinding for 20 s, followed by 30 s pauses. Samples were centrifuged for 40 s at 16,000× *g*. The resulting supernatants were transferred into Eppendorf tubes and heated for 10 min at 99 °C. Samples were subjected to a short SDS-PAGE migration and processed as previously described [28]. Briefly, the whole proteome was extracted as a single polyacrylamide band, reduced with dithiothreitol, treated with iodoacetamide and proteolyzed with sequencing-grade trypsin (Roche, Basel, Switzerland) in the presence of 0.01% of proteaseMAX detergent (Promega, Madison, WI, USA). The resulting peptides were analyzed with an ESI-Q Exactive HF mass spectrometer (Thermo Fisher Scientific, Waltham, MA, USA) coupled to an Ultimate 3000 176 RSL Nano LC System (Thermo). A volume of 10 µL of peptides was injected onto a reverse-phase Acclaim PepMap 100 C18 column (3 µm, 100 Å, 75 µm id × 500 mm) and resolved at a flow rate of 0.2 µL/min with a 60 min gradient of CH_3_CN (2.5% to 40%) in the presence of 0.1% HCOOH. The tandem mass spectrometer was operated with a Top20 strategy in data-dependent mode. Only peptide molecular ions with double or triple positive charges were selected for fragmentation with a dynamic exclusion of 10 s as previously described [29]. Tandem mass spectrometry spectra were interpreted using the MASCOT 2.2.04 software (Matrix Science, London, UK) with standard parameters. Proteins were quantified based on their spectral counts, and their abundances were normalized with total spectral counts of all proteins identified as belonging to *Frankia* for comparing distinct conditions. Proteome comparison between conditions was done according to the TFold module from the PatternLab software (www.patternlabforproteomics.org/; last access 7 January 2022).

### 2.3. Proteome Data

The mass spectrometry proteomics data were deposited in the ProteomeXchange Consortium via the PRIDE partner repository (www.ebi.ac.uk/pride/archive/; last access 16 December 2021) with the dataset identifier PXD030468 and Project DOI 10.6019/PXD030468. 

## 3. Results

The three biological replicates of symbiotic *Frankia alni* overproduced at a fold change of ≥2250 proteins (Appendix A) using a nitrogen-replete propionate-fed pure culture as reference, of which 100 had an FC ≥ 4.38 (Table 1). These 100 proteins signing for the specificity of symbiotic state account for 17.9% of the total *Frankia* proteins in nodules, based on their cumulated normalized spectral abundance factors. Conversely, there were 1489 under-produced proteins at an FC ≤ 0.5. NifH (FRAAL6813) was the most overabundant protein in this condition with an FC of 291.3.

The most underabundant was a cAMP-binding membrane-bound transcriptional regulator (FRAAL6506) at an FC of 0.01 (Appendix A) in synton (set of genes with a conserved order) with a geosmin synthase gene. There were 2983 proteins detected that belonged to *Frankia alni*. There were also 68 overabundant proteins in nodules and 30 overabundant ones in pure cultures that did not meet the *p*-value criterion (≤0.05). There were 729 proteins that were not certified by mass spectrometry with the validation of at least two distinct peptides in this dataset. The overabundant protein coding genes were scattered around the genome (Figure 1).

Among *F. alni* proteins, the nitrogenase proteins were the most overabundant with 7 among the 10 highest using as reference a nitrogen-replete pure culture. In addition to NifH, NifK (FRAAL6811) at an FC of 279 and NifD (FRAAL6812) at an FC of 86.17, the *nif* cluster comprises 17 genes from FRAAL6798 (*korB*) to FRAAL6814 (*nifV*), 13 of which were identified accounting for 3% of the cumulated NSAF in nodules.

Proteins associated to symbiotic life such as the Suf cluster (FRAAL4558-FRAAL4564, FC = 5.4 − 3.39) and the squalene lipids cluster (FRAAL1427-FRAAL1435, FC = 8.4) were among those detected as overproduced; however, the Hup clusters were not detected (FRAAL1822-FRAAL1832; FRAAL2388-FRAAL2402). Proteins involved in the upstream non-mevalonate pathway for hopanoids biosynthesis such as Dxs (FRAAL2088, FC = 2.34), Dxr (FRAAL5774, FC = 10.54), IdsA (FRAAL1431, FC = 7.75) and Idi (FRAAL6504, FC = 2.21) were also overproduced.

Other lipid metabolism proteins were overabundant such as several unlinked acetyl-CoA acetyl transferases (FRAAL4934, FC = 12.91; FRAAL2618, FC = 11.83; FRAAL0409; FC = 4.53; 3973, FC = 2.95), a fatty acid-CoA ligase (FRAAL3361, FC = 7.43), an acyl-CoA dehydrogenase (FRAAL2835, FC = 6.71) and a diacylglycerol O-transferase (FRAAL3613, FC = 5.67).

Oxidative stress proteins such as SodF (FRAAL4337, FC = 3.85) and its bacterioferritin neighbor (FRAAL4338, FC = 8.49), glutathione peroxidase (FRAAL1783, FC = 3.55), two nitroreductases (FRAAL4025, FC = 44.33; FRAAL3236, FC = 18.83) and rubrerythrin (FRAAL5116, FC = 3.08) were overabundant. Other stress proteins were underabundant such as KatA (FRAAL3167, FC = 0.09) and glutathione synthase (FRAAL4656, FC = 0.03) or were not identified such as KatB (FRAAL3889).

Several TCA cycle proteins such as succinyl-CoA synthase (FRAAL1156, FC = 6.14), aconitate hydratase (FRAAL2064, FC = 5.44), malate dehydrogenase (FRAAL5612, FC = 4.37) or fumarase (FRAAL6152, FC = 3.96) were more abundant in nodules, but several others were not detected. There were 15 ribosomal proteins overabundant at an FC > 2 (FRAAL1069, FRAAL1068, FRAAL1097, etc.) as well as other ancillary proteins such as a ribosome-recycling factor (FRAAL5778, FC = 3.35).

Other carbohydrate metabolism proteins were overabundant such as glycosyl transferases (FRAAL2049, FC = 15.5; FRAAL4552, FC = 3.5; FRAAL5400, FC = 2.85), a sugar epimerase (FRAAL6795, FC = 10.67), a ribose isomerase (FRAAL1862, FC = 9.07), a sugar-binding protein (FRAAL5914, FC = 3.49) and an enolase (FRAAL6233, FC = 3.46).

Ammonia assimilation proteins were overabundant such as the nitrogen regulatory protein P-II (FRAAL0904, FC = 3.27), glutamine-synthetase adenylyltransferase (FRAAL5164; FC = 2.13), glutamate-1-semialdehyde aminotransferase FRAAL0997, FC = 5.93) and a GS (FRAAL6426; FC = 1.8), but several others such as GSII were not detected or were underabundant such as GSI (FRAAL5161, FC = 0.01) and GOGAT (FRAAL4964-65, FC = 0.08 − 0.05).

A cell-division control ATPase was overabundant FRAAL5516 (FC = 89), while other proteins with related functions were less abundant (FRAAL6588, cell division protein FtsH FC = 0.22) or not detected.

Chaperones were overabundant such as a disaggregation chaperone (FRAAL2601, FC = 4.22) and several GroEL homologs (FRAAL6640, FC = 2.64; FRAAL1699, FC = 2.16; FRAAL1133, FC = 2.2; FRAAL6486, FC = 1.94). Other overabundant proteins with related functions were FRAAL4269 (Clp Protease), FRAAL6701 (GroEL) and FRAAL5348 (cold shock), with the corresponding genes scattered on the chromosome.

Some transporters were overabundant in nodules such as a molybdenum transporter (FRAAL1673, FC = 64.17), a BCA transporter (FRAAL3827, FC = 5.18) and a C4-dicarboxylate transporter (FRAAL1390, FC = 3.42). A phosphate transporter (FRAAL6535) was also overabundant (FC = 2.09), while many were underabundant.

Some regulators were overabundant in nodules such as a Tet-R (FRAAL0868, FC = 33.09) and a Sarp-R (FRAAL0301, FC = 12.38). A Bhl-R (FRAAL5827, FC = 9.76), a MarR (FRAAL0711, FC = 8.76) and a luxR (FRAAL4898, FC = 7.67) were also overabundant, while many were underabundant. An anti-sigma factor (FRAAL5991, FC = 5.12) in synton with a sporulation-specific sigma factor was overabundant.

Most secondary metabolite synthesis proteins were not detected or underabundant except two PKS (FRAAL5096, FC = 7.22; FRAAL4072, FC = 4.56) and a bacteriocin (FRAAL4503, FC = 2.16).

A gamma-aminobutyraldehyde dehydrogenase (FRAAL6022, FC = 3.73) was overabundant. This enzyme transforms 4-aminobutanal into 4-aminobutanoate (GABA).

The Kdp proteins (FRAAL5299-FRAAL5305), the cellulase synton proteins (FRAAL4954-FRAAL4959) and the acyl-dehydrogenases (FRAAL1659-FRAAL1670) were not identified.

Almost all carboxylases were underabundant such as three acetyl-CoA carboxylases (FRAAL3159, FC = 0.55; FRAAL1213, FC = 0.43; FRAAL3158, FC = 0.23), a methyl-crotonoyl-CoA carboxylase (FRAAL2343, FC = 0.15) and two propionyl-CoA carboxylases (FRAAL5672, FC = 0.05; FRAAL1210, FC = 0.68). The only overabundant one was a phosphoribosylaminoimidazole carboxylase (FRAAL6664, FC = 4.9) involved in the synthesis of purines.

A COG analysis (Figure 2) revealed that “C” (energy production) and “J” (translation) were the most represented in the nodule overabundant proteins, while “E” (amino acid transport and metabolism) and “K” (transcription) were the most represented in the underabundant proteins (Figure 2). Many COGs were very rare in the nodule (N, cell motility; U, intracellular trafficking; V, defense mechanisms and Q, secondary metabolites).

A drawing with the major overabundant proteins in nodules permits to show nitrogenase and other energy-generating, transporter and stress-coping determinants (Figure 3).

## 4. Discussion

Proteomics has the potential to decipher the physiological changes occurring in cells upon ecological transitions [32]. Establishment of the actinorhizal symbiosis is a drastic change for both partners that cannot be analyzed through negative genetics. Omics thus offer an interesting approach to pinpoint the key molecular players from both organisms. Proteomics in particular has been used to study *Frankia* either in early interaction with *Alnus* [18]—or with *Elaeagnus angustifolia*, *Ceanothus thyrsiflorus* and *Coriaria myrtifolia* [17], showing, among others, cellulose synthase and a potassium transporter over-detected upon contact with *Alnus* [18] and nitrogen fixation and assimilation proteins upon contact with *Elaeagnus* [17], as well as stress response and respiration proteins—or in the mature field nodules of a range of actinorhizal plants [21].

Nitrogenase is the most abundant protein complex in symbiotic *Frankia alni*. It comprises 17 genes that had previously been shown to be the most transcribed in mature nodules [7]. The present study confirms our vision that symbiotic *Frankia* is essentially a nitrogenase machine. Associated proteins were seen here, such as the SuF cluster assembly proteins and the hopanoid biosynthesis proteins (that protect nitrogenase from oxygen), but not the hydrogenase proteins even though they were among those highly expressed [33]. *Frankia* hydrogenase proteins seen previously in field nodules [21] were not detected.

The high overabundance for 2-oxoglutarate ferredoxin oxidoreductase (KorAB) confirms a role initially hypothesized based on its position next to *nif* genes of 2-oxoglutarate as a primary electron source for nitrogenase [21]. The position of the *korAB* genes next to *nif* genes and their duplication in symbiotic strains only are probably the result of a specialization on the one hand for symbiosis (FRAAL6798-6790, FC = 91.5) and on the other hand for saprophytic life (FRAAL1050-1051, FC = 0.80) as seen before for Hup [34].

Nitrogen fixation is an energy-demanding process that consumes eight ATP molecules per molecule of NH_4_^+^ produced. It is, thus, expected that the TCA cycle will be running at full force, transforming the photosynthates given by the plant into energy. The transcriptome of symbiotic *F. alni* was seen to contain several TCA genes upregulated at a high level [7] as was the case for the transcriptome of *F. soli* upon early contact [17].

There is an important specialization in nodules with only a twentieth of the genome identified (250 proteins with an FC ≥ 2) relative to a pure culture where five times as many proteins were identified at the same threshold. A similar specialization was noted in *S. meliloti* nodules [35].

The oxidative stress proteins catalase and superoxide dismutase are essential for the nodulation of *Sinorhizobium* [36] to cope with the oxidative burst induced as an early plant defense response against avirulent pathogens [37]. Nitroreductases play a similar role in *Sinorhizobium* [38]. In actinorhizal nodules, there is also an oxidative burst with which *Frankia* must cope [39]. Pure cultures of *F. alni* have been shown to have a basic expression of its two catalases that are then markedly upregulated upon contact with H_2_O_2_ or with methyl viologen [40]. However, it appears that this upregulation is not maintained in mature nodules.

GABA is considered a nutrient as well as an effector to trigger plant responses to heat, salt, herbivory and other stresses [41]. GABA has been detected in the metabolome of *Alnus* and *Casuarina* nodules and roots at very high levels [42,43]. GABA was also seen to improve nitrogenase and respiration in pure culture [43]. In leguminous nodules, GABA has been suggested to be a nutrient fed to the symbiont since it accumulated labeled ^15^N_2_ and contributed to the N-nutrition [44]. In *Alnus*, labeled ^15^NH_4_^+^ was recovered as alanine (Ala), γ-amino butyrate GABA), glutamine (Gln), glutamate (Glu), citrulline (Cit) and arginine (Arg) [45]. GABA was late to appear, but then, it continued at a fast pace. Since GABA synthesis involves the utilization of protons and releases CO_2_, it has been suggested as a means to reduce acidity [46].

The photosynthates fed by the plant to the bacterium have been the subject of several hypotheses. It has long been known that *Alnus*-infective strains could not use sugars [3], which is why organic acids are routinely used to grow most strains. The identification of a dicarboxylate transporter among those genes specific to nodules [47] indicated that dicarboxylates were good candidates. Several of these were measured in nodules and roots, and fumarate, succinate, malate and 2-oxoglutarate were found to be overabundant in nodules, although this approach is biased by the presence of mitochondria where citrate is also very abundant, yet it is toxic to *Frankia* [48]. The overabundance in the present study of the C4-dicarboxylate transporter, FRAAL1390 confirms that dicarboxylates play a role in trophic exchanges between symbionts, although its modest overabundance (FC = 3.42) points either to other transporters or to other photosynthates involved.

Chaperones such as GroEL have been shown to be essential to nodulation in *Sinorhizobium meliloti* [49]. They are also involved in response to other stresses such as heat [50] or salt [51] or simply for the assembly of very complex proteins such as nitrogenase [52]. Such chaperones were previously identified in the transcriptome of symbiotic *Frankia alni* [7].

Proteomics permits complementation of the vision achieved by transcriptomics. The two approaches have different strengths and weaknesses. Shotgun proteomics carried out in the standard mode (i.e., based on trypsin proteolysis) may miss small proteins, membrane proteins and polypeptides with rare combinations of arginine and lysine residues [53]. Transcriptomics misses unstable messengers or those with secondary structures. Furthermore, proteomics and transcriptomics address molecular entities with different half-lives and dynamics, thus they are highly complementary. The present study reinforces the picture of a microbe highly specialized for the energy-demanding nitrogenase and the accompanying stresses. It sheds no light, however, on the molecular processes involved in communication between the two partners.

## Figures and Tables

**Figure 1 microorganisms-10-00651-f001:**
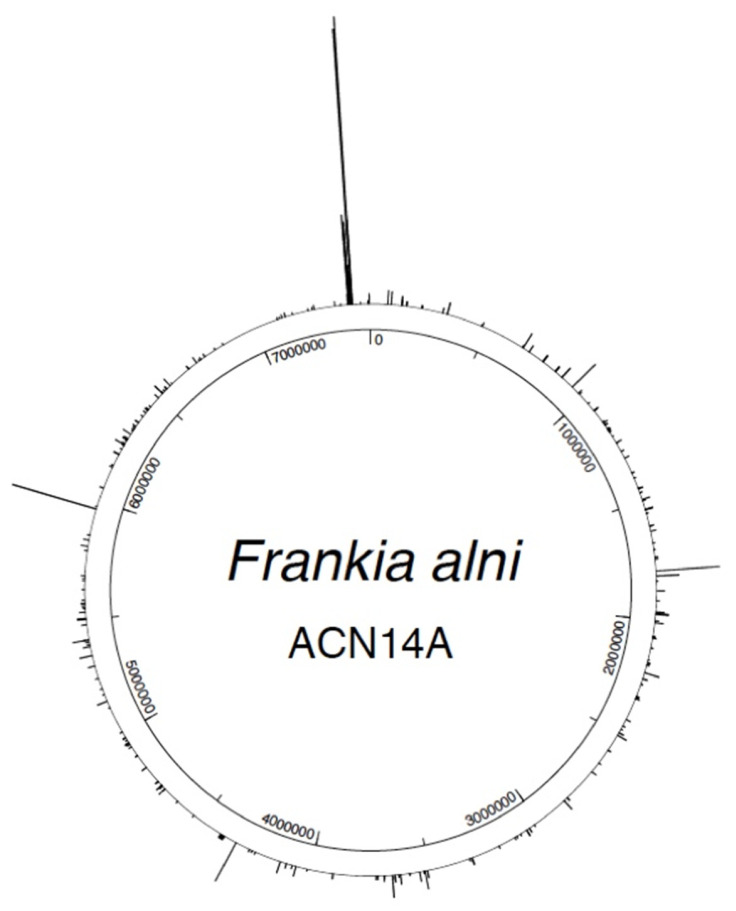
Circular map of the genome of *Frankia alni* with the proteins over-abundant in nodules relative to a nitrogen-replete pure-culture (FC ≥ 2) positioned along the genome. The *nif* genes are at the top close to the origin of replication.

**Figure 2 microorganisms-10-00651-f002:**
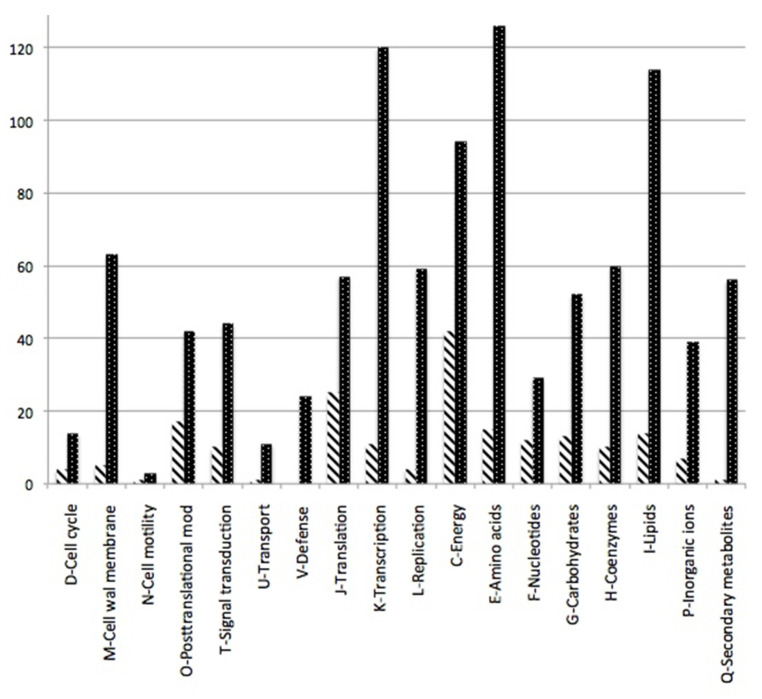
Histogram of over- and underabundant proteins coding genes (FC > 2 hatched; FC < 0.5 dotted) assigned to COGs. Undefined categories and those with one or less occurrences were omitted.

**Figure 3 microorganisms-10-00651-f003:**
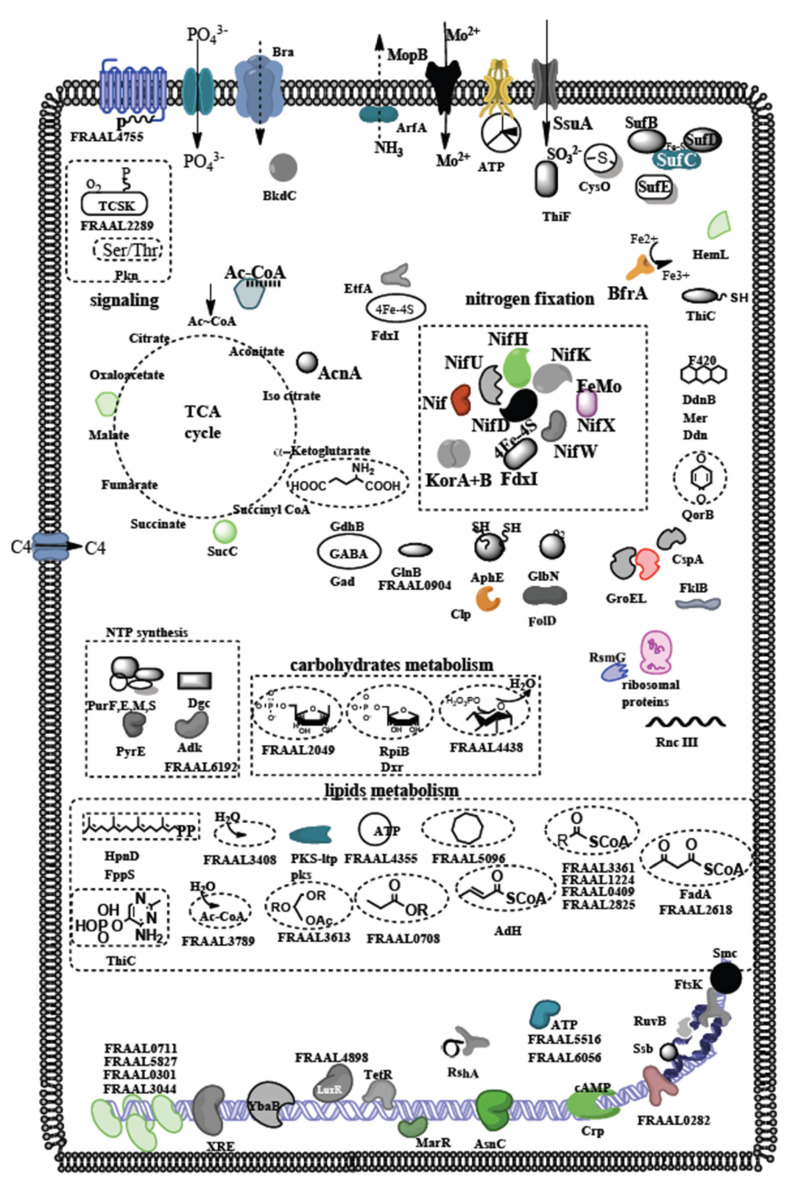
Schematics of a *Frankia* cell in its symbiotic host cell showing the likely function and localization of the 100 most abundant proteins related to nitrogen fixation, lipid metabolism, Krebs cycle (TCA), cell signaling, electron transfer, transport of anions and cations, enzymes involved in nucleotides biosynthesis, DNA metabolism and regulation and storage and stress proteins. The identities of the proteins and specific genes are listed in Table 1.

**Table 1 microorganisms-10-00651-t001:** List of the 100 most over-abundant *Frankia* proteins in nodules with the NCBI accession, the FRAAL Id, gene names, the protein name, the fold change, the COG (cluster of orthologous genes), the phylogenetic clusters distribution and the NSAF (normalized spectral abundance factors). The COGs are according to [30]. The distribution code A is present in all *Frankia* and in other actinomycetes, F is present in all *Frankia* strains, f is present in some *Frankia* strains and S is present in symbiotic *Frankia* strains (C1, C1c, C2 and C3 but not in C4). C1, C1c, C2, C3 and C4 indicate it is present in this cluster based on a threshold of 35% Id as seen on Mage [31]. Hypothetical genes have been removed from the list; they are listed in Appendix A.

NCBI Accession	FRAAL #	Gene	Functional Description	Fold Change	COG	Distribution	% NSAF Nodule
WP_009741523.1	FRAAL6813	*nifH*	Nitrogenase iron protein NifH	291.33	P	S	0.009
WP_011607845.1	FRAAL6811	*nifK*	Nitrogenase molybdenum–iron protein subunit beta NifK	279.00	C	S	0.005
WP_011607832.1	FRAAL6798	*korB*	2-Oxoglutarate ferredoxin oxidoreductase subunit beta KorB	91.50	C	S	0.002
WP_011606600.1	FRAAL5516		ATPase	89.00	O	C1	0.003
WP_011607846.1	FRAAL6812	*nifD*	Nitrogenase molybdenum–iron protein subunit alpha NifD	86.17	C	S	0.002
WP_041939983.1	FRAAL6802	*nifU*	Iron–sulfur cluster assembly accessory protein NifU	84.17	O	S	0.004
WP_041939982.1	FRAAL6799	*korA*	2-Oxoglutarate ferredoxin oxidoreductase subunit alpha KorA	82.00	C	S	0.001
WP_011602860.1	FRAAL1673	*mopB*	Molybdenum-binding protein MopB	64.17	R	S	0.004
WP_011605156.1	FRAAL4025	*ddnB*	F420H(2)-dependent quinone nitroreductase, DdnB	44.33	J	S	0.003
WP_011607839.1	FRAAL6805	*nifW*	Nitrogenase-stabilizing/protective protein NifW	40.33	P	S	0.003
WP_041939985.1	FRAAL6808	*nifX*	Nitrogen fixation protein NifX	33.83	-	S	0.002
WP_011602101.1	FRAAL0868	*tetR*	TetR family transcriptional regulator	33.09	K	C1	0.002
WP_011607841.1	FRAAL6807	*nif*	Associated nitrogen fixation protein NifX	23.83	-	S	0.003
WP_011604521.1	FRAAL3374	*mer*	Coenzyme F420-dependent N(5),N(10)-methylenetetrahydromethanopterin reductase	22.83	C	f	0.001
WP_011604385.1	FRAAL3236	*ddn*	Deazaflavin-dependent nitroreductase	18.83	-	C1	0.001
WP_011601860.1	FRAAL0611	*ybaB*	Nucleoid-associated protein	17.67	O	A	0.001
WP_011603220.1	FRAAL2049		Glycosyl transferase	15.50	M	C1c, C2	0.000
WP_011601350.1	FRAAL0082	*arfA*	Ammonia release factor ArfA	13.33	M	f	0.001
WP_011606048.1	FRAAL4934	*fadA*	3-Ketoacyl-CoA thiolase	12.91	I	A	0.003
WP_011602036.1	FRAAL0800	*thiC*	Phosphomethylpyrimidine synthase	12.80	H	A	0.000
WP_041939400.1	FRAAL3855	*fdxI*	4Fe-4S iron–sulfur binding ferredoxin	12.67	C	A	0.001
WP_011601559.1	FRAAL0301		Transcriptional regulator	12.38	T	c1	0.000
WP_011603772.1	FRAAL2618		Acetyl-CoA acetyltransferase	11.83	I	A	0.000
WP_011607074.1	FRAAL6022	*gad*	Gamma-aminobutyraldehyde dehydrogenase	11.56	C	A	0.000
WP_011604927.1	FRAAL3789		Thioesterase	10.88	R	C1, C3, C4	0.001
WP_011603462.1	FRAAL2298	*folD*	Methylenetetrahydrofolate dehydrogenase FolD	10.86	H	A	0.001
WP_011605870.1	FRAAL4755		Membrane phosphatase	10.80	K	A	0.000
WP_011607913.1	FRAAL6878	*rsmG*	Ribosomal RNA small subunit methyltransferase G	10.77	M	A	0.001
WP_011607828.1	FRAAL6795	*qorB*	Quinone oxydoreductase QorB	10.67	M	C1, C3, C4	0.001
WP_011606846.1	FRAAL5774	*dxr*	1-Deoxy-D-xylulose 5-phosphate reductoisomerase Dxr	10.54	I	A	0.001
WP_050997128.1	FRAAL3342	*ssuA*	Sulfonate transport system substrate-binding SsuA	10.50	P	C1, C2, C4	0.000
WP_041938619.1	FRAAL0128	*purF*	Amidophosphoribosyltransferase PurF	9.86	F	A	0.000
WP_009742300.1	FRAAL5827		DNA-binding protein HU-beta, NS1 (HU-1)	9.76	L	A	0.016
WP_041939818.1	FRAAL6056		ATP-binding protein	9.60	-	A	0.004
WP_041939008.1	FRAAL1862	*rpiB*	Ribose 5-phosphate isomerase RplB	9.07	G	A	0.003
WP_011601953.1	FRAAL0711		Transcriptional regulator	8.76	K	C1, C1c, C3, C4	0.002
WP_009737569.1	FRAAL4338	*bfrA*	Bacterioferritin BfrA	8.49	P	A	0.004
WP_011602931.1	FRAAL1747		Putative carboxyvinyl–carboxyphosphonate phosphorylmutase	8.43	G	C1, C1c, C4	0.000
WP_041938916.1	FRAAL1429	*hpnD*	Phytoene synthase HpnD	8.40	I	A	0.003
WP_011604375.1	FRAAL3226	*bkdC*	Branched-chain alpha-keto acid dehydrogenase subunit BkdC	8.14	C	A	0.000
WP_041940119.1	FRAAL0708		Lipid esterase	7.91	R	C1, C2, C3, C4	0.000
WP_011602626.1	FRAAL1431	*fppS*	Farnesyl diphosphate synthase	7.75	H	A	0.002
WP_041939587.1	FRAAL4898		LuxR family transcriptional regulator	7.67	K	C1, C1c, C3, C4	0.000
WP_011606077.1	FRAAL4963	*adk*	Adenylate kinase	7.50	F	A	0.001
WP_011604508.1	FRAAL3361		Short-chain fatty acid–CoA ligase	7.43	I	A	0.000
WP_011604193.1	FRAAL3044		Transcriptional regulator	7.36	K	f	0.000
WP_041939680.1	FRAAL5348	*cspA*	Cold shock protein	7.33	K	A	0.002
WP_011602011.1	FRAAL0771	*dgc*	Diguanylate cyclase	7.17	T	f	0.000
WP_011607242.1	FRAAL6192		Adenylate kinase	7.17	F	A	0.001
WP_041939231.1	FRAAL2835	*adh*	Acyl-CoA dehydrogenase	6.71	I	A	0.000
WP_011606204.1	FRAAL5096	*pks*	Polyketide cyclase/dehydrase and lipid transport	6.56	-	A	0.001
WP_050997247.1	FRAAL5736	*ftsK*	Cell division protein FtsK	6.43	D	A	0.000
WP_011604554.1	FRAAL3408		Alpha/beta hydrolase	6.37	R	f	0.001
WP_041938863.1	FRAAL1156	*sucC*	Succinyl-CoA synthetase subunit beta SucC	6.14	C	A	0.012
WP_011605559.1	FRAAL4438		Sugar-phosphate dehydrogenase	6.08	C	A	0.002
WP_041939760.1	FRAAL5796	*smc*	Chromosome segregation protein Smc	6.04	D	A	0.000
WP_041939603.1	FRAAL4991	*thiF*	Thiamine biosynthesis protein ThiF	6.00	H	f	0.000
WP_041938832.1	FRAAL0997	*hemL*	Glutamate-1-semialdehyde aminotransferase HemL	5.93	H	A	0.001
WP_011603632.1	FRAAL2475	*glbN*	Truncated hemoglobin GlbN	5.87	R	f	0.001
WP_011601393.1	FRAAL0129	*purM*	Phosphoribosylaminoimidazole synthetase PurM	5.77	F	A	0.001
WP_041938655.1	FRAAL0282		Putative integrase/resolvase	5.71	L	f	0.000
WP_011603454.1	FRAAL2289		Putative two-component oxygen sensor kinase	5.67	T	A	0.000
WP_011604752.1	FRAAL3613		Diacylglycerol O-acyltransferase	5.67	-	f	0.000
WP_011602534.1	FRAAL1337	*pkn*	Serine/threonine protein kinase	5.50	T	F	0.000
WP_011603231.1	FRAAL2064	*acnA*	Aconitate hydratase AcnA	5.44	C	A	0.006
WP_011605678.1	FRAAL4561	*sufE*	(2Fe-2S)-Binding protein SufE	5.40	P	A	0.001
WP_050997107.1	FRAAL2825		Putative N-fatty-acyl-amino acid synthase	5.38	E	A	0.001
WP_011606156.1	FRAAL5045	*ahpE*	Peroxiredoxin AhpE	5.31	O	A	0.009
WP_041939398.1	FRAAL3827	*bra*	Branched-chain amino acid ABC transporter ATP-binding protein	5.18	E	A	0.001
WP_011607046.1	FRAAL5991	*rshA*	Anti-sigma factor RshA	5.12	T	A	0.004
WP_011602689.1	FRAAL1499	*cysO*	Sulfur carrier protein CysO	5.07	H	A	0.002
WP_011602425.1	FRAAL1224		Acetyl-CoA synthetase	5.04	I	A	0.001
WP_011602282.1	FRAAL1055	*pyrE*	Orotate phosphoribosyltransferase pyrE	5.00	F	A	0.001
WP_041939967.1	FRAAL6738	*purS*	Phosphoribosylformylglycinamidine synthase PurS	4.92	F	f	0.001
WP_041939945.1	FRAAL6664	*purE*	Phosphoribosylaminoimidazole carboxylase PurE	4.90	F	A	0.003
WP_035923438.1	FRAAL6556	*crp*	Crp/Fnr family transcriptional regulator	4.83	T	A	0.005
WP_011606876.1	FRAAL5804	*rnc*	Ribonuclease III rnc	4.82	K	A	0.000
WP_011604517.1	FRAAL3370	*fklB*	Peptidyl–prolyl cis–trans isomerase FklB	4.80	O	A	0.003
WP_041939534.1	FRAAL4560	*sufC*	Fe–S cluster assembly ATPase SufC	4.62	O	A	0.004
WP_011603306.1	FRAAL2141	*ruvB*	Holliday junction DNA helicase RuvB	4.60	L	A	0.000
WP_011605202.1	FRAAL4072	*pks*	Polyketide synthase	4.56	Q	f	0.000
WP_011601665.1	FRAAL0409		Acetyl-CoA acetyltransferase	4.53	I	A	0.000
WP_011606938.1	FRAAL5877	*etfA*	Electron transfer flavoprotein EtfA	4.48	C	A	0.007
WP_011601417.1	FRAAL0156	*gdhB*	Glutamate dehydrogenase GdhB	4.47	E	f	0.000

## Data Availability

The mass spectrometry proteomics data were deposited in the ProteomeXchange Consortium via the PRIDE partner repository (www.ebi.ac.uk/pride/archive/; 16 December 2021) with the dataset identifier PXD030468 and Project DOI 10.6019/PXD030468.

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
