# Peer review of "The Proteogenome of Symbiotic Frankia alni in Alnus glutinosa Nodules"

_microorganisms, 2022, doi:10.3390/microorganisms10030651_

Round 1

Reviewer 1 Report

The symbiotic nitrogen fixing symbiosis between Frankia and its plant host, in this case Alnus,  is of considerable interest in the biological nitrogen fixation field and has been studied for many years by the principle investigator of the laboratory of the Authors.  There are significant differences and also similarities between the Frankia-Alnus system and the widely studied symbiotic nitrogen fixing symbiosis between rhizobia and their legume host (e.g. the Sinorhizobium-Medicago system), meriting a closer look at the biology of the former.  One major difference between these symbiotic systems is the absence of a genetic system for the Frankia-Alnus symbiosis, which has slowed down progress in deciphering the signalling between the symbiotic partners and other aspects of the infection, nodule ontology and nitrogen fixation processes.  Therefore genomic, transcriptomics and proteomics approaches have been carried out to try to  elucidate the Frankia-Alnus interaction.  Here a Proteomics comparison of symbiotic Frankia and in vitro nitrogen fixing cells is presented.  Approximately 100 proteins were foun to be overproduced in symbiotic Frankia cells, including several proteins previously identified by omics approaches, allowing the Authors to draw a schematic representation of the presumptive physiological parameters of a Frankia cell in the symbiotic state.  Nitrogenase, N-assimilation proteins, transporters, TCA cycle enzymes and oxidative stress response proteins were some of the proteins identified, which are to be expected in a nitrogen fixing cell, but a relatively small proportion of novel Frankia proteins were found.  The latter applies especially to signalling and infection components.  The study was carefully crafted and well described and helped to envision the infected cell as a "nitrogen fixation machine".  However few novel insights into the biology of the Frankia-Alnus symbiotic system were revealed.  Therefore this paper Is of considerable interest to the Frankia research community, but of average significance to the larger research community. Nevertheless I recommend publication of this extensive research project in Microbiology.

Author Response

The reviewer has not asked for any modification to the paper.

We thank her/him for her/his positive comments

Reviewer 2 Report

Pujic et al compared the proteome of pure cultures of Frankia with 21 dpi Alnus glutinosa nodules. The study is only descriptive, but it provides a good basis for future studies on the interaction between Frankia and its host plant.

There are number of minor points to be addressed:

Please explain all abbreviations when they are used first, and don't use abbreviations in the abstract such as FC, Sod, SuF.

Please briefly mention the function of nitrogenase and hopanoid lipids in the introduction. Not everybody is familiar with the role of these proteins.

Table 1 is far too long for the main part of the manuscript. Please shorten it and it might be helpful to group the proteins found based on key functions (e.g. nitrogen fixation, lipid metabolism etc.). The full table can be shown in Supplementals. And again, please explain abbreviations such as NSAF, COG etc.

Line 136 What do you mean by in synton?

Figure 2: please label the x-axis properly. It is not very helpful to find the labeling only in the legend. The writing can be in vertical order and adapt the text (lines 214-218) accordingly. You may use a bar graph instead of columns.

line 238 lipid metabolism

line 245 was (not were)

line 260 of (not for)

line 262 nitrogenase (not N2ase)

line 262 what is the reference for "their"? 2-oxoglutarate ferridoxin oxidoreductase? If yes please use its position... and its....

line 263 are evocative  - I cannot follow, what do you want to mean here?

line 275 catalase and superoxide dismutase

line 278 Pure cultures of F. alni have been

line 281 However, it appears

lilne 308 achieved (not afforded)

Author Response

Please explain all abbreviations when they are used first, and don't use abbreviations in the abstract such as FC, Sod, SuF.

                  All abbreviations are explained upon first use.

Please briefly mention the function of nitrogenase and hopanoid lipidsin the introduction. Not everybody is familiar with the role of these proteins.

                  The functions of N2ase and hopaoid lipids are given in the introduction

Table 1 is far too longfor the main part of the manuscript. Please shorten it and it might be helpful to group the proteins found based on key functions (e.g. nitrogen fixation, lipid metabolism etc.). The full table can be shown in Supplementals. And again, please explain abbreviations such as NSAF, COG etc.

                  Table1 has been shortened by removing two columns (MW, pI) and 16 rows (all hypothetical proteins). Abbreviations are explained.

Line 136 What do you mean by in synton?

            A synton is a set of homologous genes which relative grouping is conserved on the genome, this has been defined the first time the name was used.

Figure 2: please label the x-axis properly. It is not very helpful to find the labeling only in the legend. The writing can be in vertical order and adapt the text (lines 214-218) accordingly. You may use a bar graph instead of columns.

                  Horizontal axis has been labeled as required. Legend was also modified.

line 238 lipid metabolism

            Modified as suggested

line 245 was (not were)

                  The subject has been modified ("hydrogenase" for "hydrogenase proteins" so the verb remain plural ("were").

line 260 of (not for)

                  Modified as suggested

line 262 nitrogenase (not N2ase)

                  Modified as suggested

line 262 what is the reference for "their"? 2-oxoglutarate ferridoxin oxidoreductase? If yes please use its position... and its....

                  Text modified to specify we talk about the kor genes

line 263 are evocative  - I cannot follow, what do you want to mean here?

                  We mean the conserved position of the kor genes next to the nif cluster and their duplication are probably the result of a need for non-limiting nitrogenase function of a large supply of 2-oxoglutarate. Wording has been modified.

line 275 catalase and superoxide dismutase

            Modified as suggested

line 278 Pure cultures of F. alni have been

                  Modified as suggested

line 281 However, it appears

                  Modified as suggested

lilne 308 achieved (not afforded)

                  Modified as suggested